# Comparative Analysis of Phytochemical and Functional Profiles of Arabica Coffee Leaves and Green Beans Across Different Cultivars

**DOI:** 10.3390/foods13233744

**Published:** 2024-11-22

**Authors:** Yoon A Jeon, Premkumar Natraj, Seong Cheol Kim, Joon-Kwan Moon, Young Jae Lee

**Affiliations:** 1College of Veterinary Medicine, Jeju National University, Jeju 63243, Republic of Korea; yoonaj0915@naver.com (Y.A.J.); premkumar.n@jejunu.ac.kr (P.N.); 2Research Institute of Climate Change and Agriculture, National Institute of Horticultural and Herbal Science, Rural Development Administration, Jeju 63240, Republic of Korea; kimsec@korea.kr; 3Department of Plant Life and Environmental Sciences, Hankyong National University, Anseong 17579, Republic of Korea; jkmoon@hknu.ac.kr; 4Veterinary Medical Research Institute, Jeju National University, Jeju 63243, Republic of Korea

**Keywords:** chlorogenic acids, caffeine, 2,2-Diphenyl-1-picrylhydrazyl, flavonoids, nuclear factor kappa B, nitric oxide

## Abstract

This study analyzed the phytochemical composition and functional properties of leaves and green beans from seven Arabica coffee cultivars. The total phenolic and flavonoid contents were measured using spectrophotometric methods, while caffeine, chlorogenic acid (CGA), and mangiferin levels were quantified via High-Performance Liquid Chromatography (HPLC). Volatile compounds were identified using Gas Chromatography–Mass Spectrometry (GC-MS). Antioxidant activity was assessed using 2,2-Diphenyl-1-Picrylhydrazyl (DPPH) and 2,2′-azino-bis(3-ethylbenzothiazoline-6-sulfonic acid) (ABTS) radical scavenging assays, and anti-inflammatory effects were evaluated by measuring reactive oxygen species (ROS), nitric oxide (NO) levels, and nuclear factor kappa B (NF-κB) activation in lipopolysaccharide (LPS)-stimulated macrophages. The results revealed that coffee leaves had significantly higher levels of total phenols, flavonoids, and CGAs, and exhibited stronger antioxidant activities compared to green beans. Notably, Geisha leaves exhibited the highest concentrations of phenolics and flavonoids, along with potent anti-inflammatory properties. Among green beans, the Marsellesa cultivar exhibited a significant flavonoid content and strong ABTS scavenging and anti-inflammatory effects. GC-MS analysis highlighted distinct volatile compound profiles between leaves and green beans, underscoring the phytochemical diversity among cultivars. Multivariate 3D principal component analysis (PCA) demonstrated clear chemical differentiation between coffee leaves and beans across cultivars, driven by key compounds such as caffeine, CGAs, and pentadecanoic acid. Hierarchical clustering further supported these findings, with dendrograms revealing distinct grouping patterns for leaves and beans, indicating cultivar-specific chemical profiles. These results underscore the significant chemical and functional diversity across Arabica cultivars, positioning coffee leaves as a promising functional alternative to green beans due to their rich phytochemical content and bioactive properties.

## 1. Introduction

Coffee is one of the most widely consumed beverages globally, with an annual production exceeding 10 million tons, primarily cultivated in tropical and subtropical regions, including Brazil, Vietnam, Colombia, and Indonesia [1,2]. The two predominant varieties of coffee are *Coffea arabica* and *Coffea canephora* (Robusta), with Arabica prized for its superior flavor profile and rich bioactive compounds, including caffeine and chlorogenic acids (CGAs) [3]. These bioactive compounds are associated with numerous health benefits, such as antioxidative, anticancer, and cardiovascular effects, making coffee not only a popular beverage but also a potential functional food [4,5,6]. The impacts of climate change have led to significant challenges in traditional coffee-growing regions, causing unstable production and economic strain on communities reliant on coffee cultivation [7,8]. In response, coffee production has expanded into temperate regions, including Japan and Korea, where cultivation under controlled greenhouse conditions aims to maintain crop quality [9]. However, there is limited information on the phytochemical profiles and functional properties of coffee beans grown in these non-traditional regions, particularly regarding their suitability for producing high-quality functional foods.

In addition to coffee beans, there is growing interest in the valorization of coffee by-products, such as coffee leaves, to mitigate environmental impacts and enhance economic benefits for farmers [1,10]. Farmers customarily remove leaves as a standard cultivation method to ensure optimal plant growth and the production of high-quality beans. Therefore, transforming coffee leaves into products aligns with sustainable practices, reducing waste and promoting eco-friendly consumption [11]. In 2020, the European Union approved coffee leaves as a tea beverage, recognizing their significant health benefits [12]. Innovative products such as BALDAKO HEALTHY provide a range of flavors, enhancing the attraction of coffee leaf beverages to customers [13]. Coffee leaves contain a diverse range of bioactive compounds, including alkaloids, flavonoids, terpenes, tannins, and phenolic acids, which are known for their antioxidant, anti-inflammatory, antihypertensive, antibacterial, and antifungal activities [14,15,16]. Notably, coffee leaves contain mangiferin, a compound absent in green beans [17], which is recognized for its potent antioxidant and anti-inflammatory properties, further expanding the functional potential of coffee leaves [18,19,20].

Despite these recognized benefits, comprehensive analyses comparing the phytochemical composition, including caffeine levels, CGAs, and other bioactive compounds, and functional properties of coffee leaves to those of green beans across different Arabica cultivars remain limited. This study aims to address this gap by evaluating the phytochemical profiles, including phenolic compounds, flavonoids, caffeine, CGAs, and mangiferin, of leaves and green beans from seven Arabica coffee cultivars (Catuai, Caturra, Costarica, Geisha, Marsellesa, Obata, and Venecia) grown in a greenhouse in Jeju, Korea. Furthermore, this study examines the antioxidant and anti-inflammatory activities of these cultivars using Diphenyl-1-Picrylhydrazyl (DPPH) and 2,2′-azino-bis(3-ethylbenzothiazoline-6-sulfonic acid) (ABTS) radical scavenging assays and bioassays measuring reactive oxygen species (ROS) production, nitric oxide (NO) levels, and nuclear factor kappa B (NF-κB) activation in lipopolysaccharide (LPS)-stimulated macrophages.

This study aims to highlight the potential of coffee leaves as a viable functional alternative to green beans by providing detailed comparative data on the phytochemical composition and functional properties of Arabica coffee leaves and green beans. The findings could inform future research and industrial applications, supporting the development of sustainable and economically viable coffee cultivation in temperate regions and expanding the utility of coffee plant components beyond traditional uses.

## 2. Materials and Methods

### 2.1. Sample Preparation

The seven Arabica coffee cultivars were grown in the greenhouse in the experimental orchard of the Research Institute of Climate Change and Agriculture, the National Institute of Horticultural and Herbal Science, Jeju, Republic of Korea (33°28′ N, 126°31′ E): Catuai, Caturra, Costarica, Geisha, Marsellesa, Obata, and Venecia. The minimum temperatures in the winter season were maintained above 10 °C to prevent coffee trees from incurring chilling or freezing injuries. Coffee leaves and green beans were harvested in June 2022. The leaves and green beans of each cultivar were dried at 20–25 °C and were ground and soaked in 80% methanol for 24 h. After filtration, the extracts were evaporated in a rotary evaporator (IKA RV8, IKA-Werke GmbH and Co. KG, Staufen im Breisgau, Germany) at 50 °C. Samples were freeze-dried and stored at −20 °C before further analyses.

### 2.2. Determination of Total Phenol and Total Flavonoid Contents

The total phenolic contents of coffee leaves and green beans were determined using the Folin–Denis method. Extracts were reacted with Folin–Ciocalteu’s phenol reagent (Sigma Aldrich, St. Louis, MO, USA) and a 7% sodium carbonate (Na_2_CO_3_, *w*/*v*) solution for 60 min. The absorbance was measured at 720 nm using a microplate reader (SpectraMax M3, Molecular Device, San Jose, CA, USA). The total phenolic content was expressed as mg of gallic acid equivalents per g of dry weight (mg GAEs/g DW).

The total flavonoid content was determined using the previously described method [21]. Briefly, extracts were reacted with a 2% aluminum chloride (AlCl_3_, *w*/*v*) solution for 15 min. The absorbance was measured at 430 nm using a microplate reader. The total flavonoid content was expressed as mg of quercetin equivalents per g of dry weight (mg QUEs/g DW).

### 2.3. Determination of Caffeine, Chlorogenic Acid, and Mangiferin Contents

Quantitative analyses of caffeine, CGAs, and mangiferin were performed using a Shimadzu NexeraXr HPLC system equipped with a Phenomenex Gemini C-18 5 μ column (250 mm × 4.6 mm i.d.) and a multiple wavelength detector. Mobile phase A was water (containing 0.1% formic acid), and mobile phase B was acetonitrile (containing 0.1% formic acid). The gradient mode was initially set at an A/B ratio of 95/5 from 0 to 3 min, then linearly increased to 75/25 from 45 to 50 min and then to 50/50 from 53 to 57 min. The flow rate was 0.8 mL/min. The detector was set at 325 nm for CGAs, 272 nm for caffeine, and 317 nm for mangiferin; the injection volume was 10 μL. Concentrations were calculated using the regression equation of their concentration and the peak area of standards as mg/g extract.

### 2.4. Determination of Radical Scavenging Activities

The radical scavenging capacity of coffee leaves and green beans was evaluated as previously described [21]. The samples were reacted with 0.2 mM 2,2-Diphenyl-1-picrylhydrazyl (DPPH, Sigma Aldrich, St. Louis, MO, USA) for 20 min at ambient temperature, and the absorbance was measured at 517 nm using a microplate reader.

A total of 7.4 mM of 2,2′-Azino-bis(3-ethylbenzothiazoline-6-sulfonic acid) (ABTS) was mixed with 2.6 mM potassium persulfate for 16 h to convert ABTS to ABTS^+^. The samples were reacted with a diluted ABTS mixture for 15 min, and the absorbance was measured at 734 nm using a microplate reader.

The radical scavenging capacity was calculated with the following equation:Radical scavenging (%) = (1 − absorbance of sample/absorbance of control) × 100

As previously described [21], the half maximal effective concentration (EC_50_) was determined by constructing a dose–response curve through non-linear regression analysis using GraphPad Prism 6.0.0 (GraphPad Software, San Diego, CA, USA).

### 2.5. Determination of Anti-Inflammatory Activities

#### 2.5.1. Cell Culture

RAW-Blue™ cells (InvivoGen, San Diego, CA, USA) were cultured in Dulbecco’s modified Eagle’s medium (Gibco, San Francisco, CA, USA), supplemented with 10% fetal bovine serum (Gibco, San Francisco, CA, USA) and 1% penicillin/streptomycin at 37 °C and 5% CO_2_. The cells were seeded in 96-well plates at a density of 3 × 10^4^ cells/mL and cultured for 24 h. Then, cells were incubated with 0–400 μg/mL of the extracts for 1 h, and further treated with 1 μg/mL of LPS.

#### 2.5.2. Cell Viabilities

The cell viability was assessed using the 3-(4,5-dimethylthiazol-2-yl)-2,5-diphenyltetrazolium bromide (MTT) assay to determine the cytotoxicity of leaves and green bean extract from seven cultivars of Arabica coffee. After treatment as described above, cells were incubated with the MTT solution for four hours, and the supernatant was discarded. The cells were de-crystallized using dimethyl sulfoxide, and the absorbance was measured at 570 nm using a microplate reader. The cell viability ratio (%) was calculated based on the control.

#### 2.5.3. Measurement of Nitric Oxide (NO) Production

The NO production in RAW-Blue™ cells was measured indirectly through the analysis of nitrite levels using the Griess assay. After treatment as described above, Griess reagent (Sigma Aldrich, St. Louis, MO, USA) was added to the cell culture media, and the absorbance was measured at 540 nm using a microplate reader. The NO production ratio (%) was calculated based on an LPS-treated control.

#### 2.5.4. Measurement of Reactive Oxygen Species (ROS) Production

The intracellular ROS level was measured using the fluorescence dye 2′,7′-dichlorodihydrofluorescein diacetate (H_2_DCF-DA, Invitrogen, Waltham, MA, USA). After treatment, cells were incubated with H_2_DCF-DA for 30 min. After washing them with PBS, the intracellular fluorescence intensity of DCF (excitation of 492–495 nm and emission of 517–527 nm) was measured using a plate reader (Synergy-HTX, Agilent, Santa Clara, CA, USA). The results were expressed as a percentage of the LPS-treated control.

#### 2.5.5. Measurement of NF-κB Activation

To measure the inhibition of NF-κB activation, the cell culture media were mixed with QUANTI-Blue™ (InvivoGen) reagent at 37 °C for four hours after the treatment described above. The absorbance was then measured at 620 nm. The inhibition of NF-κB activation was calculated relative to the values of the LPS-treated control and reported as the percentage expression of secreted embryonic alkaline phosphate (SEAP).

### 2.6. GC-MS Analysis

The methanolic extracts of coffee were analyzed using a Shimadzu QP-2010 Ultra GC-MS system (Shimadzu, Kyoto, Japan). The system was equipped with a capillary standard and a 60 M TRX 5-MS non-polar column (30 m length, 0.25 mm internal diameter, 0.25 μm film thickness). Helium was used as the carrier gas at a flow rate of 1.21 mL/min. The oven temperature was programmed to increase from 100 °C to 260 °C at a rate of 10 °C/min. A sample injection volume of 2 μL was used, with the electron ionization energy set at 70 eV. The total run time for each sample was 30 min, with extracts dissolved in methanol and analyzed over a mass range of 10–850 *m*/*z*. The mass spectra were recorded within a retention time range of 0–30 min [22]. The identification of components was performed by comparing the mass spectra with those in the Wiley spectral library and the NIST Mass Spectral Library Ver. 2.0 d, as well as with validated samples from Sigma-Aldrich. The relative proportion of each component was calculated by comparing its average peak area to the total peak area. Additional characteristics, including the molecular formula, molecular weight, and the structure of the bioactive components, were established by comparing the *m*/*z* ratios with the literature data [23].

### 2.7. Statistical Analyses

Data were analyzed using Prism 6.0.0. and presented as the mean ± SEM. Statistical analyses were performed using a non-parametric one-way ANOVA or two-way ANOVA, assuming equal variance (homoscedasticity) based on similar sample sizes and the absence of significant outliers. Tukey’s post hoc test was used to determine significant differences between groups (*p* < 0.05). Correlation coefficients were also calculated for the relationships between the phytochemical contents and function.

For volatile compound data analysis, principal component analyses (PCA) and hierarchical clustering analysis were performed using OriginPro 2024. A dendrogram was generated in OriginPro 2024, where the clustering was based on the Euclidean distance and Ward’s linkage method. The hierarchical clustering grouped the coffee samples into distinct clusters, representing the degree of similarity or dissimilarity in their phytochemical compositions. The height of the branches (linkage distance) in the dendrogram indicated the extent of the chemical differences between leaves and green beans.

## 3. Results and Discussion

### 3.1. Total Phenol and Flavonoid Contents

The total phenol concentrations (TPCs) of Arabica coffees exhibited considerable variations among cultivars and parts, as they varied between 397.1 and 1396.8 mg GAEs/g DW. Leaf extracts had a higher TPC (867.7 ± 273.29 mg GAEs/g DW) compared to green bean extracts (486.1 ± 74.91 mg GAEs/g DW). The leaf extract of Geisha (1396.8 ± 28.04 mg GAEs/g DW) and the green bean of Marsellesa (581.1 ± 13.03 mg GAEs/g DW) exhibited the highest levels of the TPC. Our findings match with previous research which found a TPC in unroasted coffee green beans ranging from 49.2 to 74 g GAEs/kg [24] and in Arabica coffee leaves ranging from 39.3 to 67.6 mg GAEs/g [25].

While research on flavonoids in coffee is relatively less developed, it is known that Arabica coffee contains catechins, kaempferol, and quercetin [3,26]. The total flavonoid contents (TFCs) also varied significantly among cultivars and parts, ranging from 0 to 70.69 mg GAEs/g DW (Figure 1B). The TFC in leaf extracts varied, with Geisha showing the highest level (70.7 ± 2.31 mg QUEs/g DW) and Obata showing the lowest (10.8 ± 0.45 mg QUEs/g DW). In green bean extracts, the TFCs of Marsellesa had the highest level (19.6 ± 2.73 mg QUEs/g DW), and the Obata cultivar showed no detectable content. Ngibad et al. [27] reported that the TFC of roasted Arabica coffee is approximately 9.16 mg QUEs/g. Another study reported that the TFC of Nepalese Arabica green beans ranges from 8 to 10 mg QUEs/mL [28]. These differences in findings highlight the need for further research into the changes in TFC in relation to roasting levels. Indeed, Jung et al. [29] reported that the TFC levels changed according to the degree of heat treatment.

The findings indicate that coffee leaves generally possess higher levels of total phenolic and flavonoid contents compared to green beans. This is consistent with previous studies that have demonstrated the rich phytochemical profile of coffee leaves [15,16]. The Geisha cultivar, in particular, exhibited the highest levels of phenols and flavonoids, suggesting its superior potential for health benefits.

### 3.2. Chlorogenic Acid, Caffeine, and Mangiferin Contents

CGAs, which are the primary polyphenolic chemicals found in coffee, are renowned for their various health advantages [30,31,32,33,34]. Table 1 displays the differences in the levels of CGAs, such as 3-caffeoylquinic acid (3-CQA), 4-CQA, 5-CQA, 4-feruloylquinic acid (4-FQA), 5-FQA, 3,5-dicaffeoylquinic acid (3,5-DiCQA), and 4,5-DiCQA, among the leaves and green beans of various Arabica coffees. Among CGAs, 5-CQA had the highest concentration, ranging from 54.5 to 324.3 mg/g extract. Similarly, a comparative study conducted on several coffee species revealed that 79% of the CGA in Arabica coffee is composed of CQAs [35]. The leaf extracts had 4.4 to 6.0 times higher 5-CQA contents than green beans. Among the cultivars, Venecia had the highest content of 3-CQA, and Marsellesa had the highest content of 5-FQA in both leaves and green beans. The Marsellesa leaf also showed the highest 5-CQA content at 324.3 ± 4.97 mg/g extract. The elevated levels of CGAs in leaves, especially 5-CQA, further support the notion that coffee leaves could be a valuable source of bioactive compounds.

The caffeine concentration in green beans was determined to be 2.2–2.6 times greater than that in leaf extracts for all cultivars (Figure 2A). Similarly, research on Ethiopian coffee found that the caffeine content in green coffee beans was at least 73% higher than in leaves [36]. The Obata cultivar showed the highest caffeine concentration, measuring 153.2 mg/g in the leaf and 350.3 mg/g in the green bean. One study indicated that the caffeine amount in green coffee beans is 83.89 µg/mg, which is significantly lower than the caffeine content in our green coffee beans [37]. Regarding coffee leaves, previous studies reported caffeine concentrations of 14.94 mg/g DW [38] and 7.94 mg/g DW [39]. These differences in the caffeine contents can be derived from the growth conditions of the coffee tree [40,41]. Indeed, Getachew et al. indicated that the elevation, soil temperatures, and soil chemical characteristics influence the caffeine contents of green beans [42].

The mangiferin content was quantified explicitly in the leaves, and the Marsellesa cultivar exhibited the greatest amount at 8.6 ± 0.13 mg/g extract (Figure 2B). Coffee leaves are notable for containing not only the primary metabolites found in seeds but also mangiferin, which is recognized for its antioxidant, antibacterial, and diverse pharmacological characteristics [41]. Prior research indicated that the mangiferin concentration in Arabica coffee leaves was 4.43 mg/g DW [39]. In contrast, the leaves of the Bourbon cultivar had a mangiferin content of 2.22–14.71 mg/g DW [38]. These differences can occur due to environmental factors, such as the altitude and ultraviolet radiation [43].

### 3.3. Radical Scavenging Activities

The leaf and green bean extracts from Arabica cultivars showed notable differences in their DPPH and ABTS radical scavenging capabilities, as depicted in Figure 3. The EC_50_ value for the DPPH radical scavenging activity was lower in leaf extracts (35.6 ± 14.13 μg/mL) compared to green bean extracts (72.9 ± 7.78 μg/mL). The EC_50_ values for the leaf extracts varied between 21.0 ± 1.11 μg/mL for Geisha and 54.5 ± 1.17 μg/mL for Catuai. The EC_50_ values for green bean extracts ranged from 64.5 ± 1.18 μg/mL for Venecia to 83.1 ± 1.12 μg/mL for Marsellesa.

Similarly, the ABTS radical scavenging ability showed differences among the cultivars. The Geisha, Obata, and Venecia cultivars had an exceptional scavenging capacity in the leaves, but Caturra revealed superior performance in green beans. Obata showed the most effective results for both leaves and green beans, with EC_50_ values of the ABTS scavenging activity of 18.6 ± 1.18 and 23.5 ± 1.11 μg/mL, respectively.

Plants, especially leaves, need systems to remove reactive oxygen species produced by UV radiation and surplus energy, resulting in the production of more antioxidants than seeds [44]. However, no notable differences were observed in the ABTS scavenging capacity between leaves and green beans. Yang et al. mentioned that the ABTS assay utilizes cationic radicals that rely on an electron transfer process, while the DPPH assay employs neutral radicals that operate through a hydrogen atom transfer mechanism [45]. The different patterns in the DPPH and ABTS radical scavenging activities can be ascribed to the mechanisms of these methods. Additionally, while it is commonly observed that the antioxidant capacity is typically higher when there are more hydroxyl substituents present, there are some cases that do not follow this pattern [46,47]. This underscores the necessity of considering differences in the contained compounds when evaluating coffee’s potential health advantages. Despite these complexities, our study is valuable as it is the first to evaluate the antioxidant activity of leaves and green beans from different cultivars grown under identical conditions.

### 3.4. Anti-Inflammatory Activities

The cell viability of LPS-treated cells with and without coffee extracts up to 400 µg/mL did not exhibit any significant differences. The RAW-Blue™ cells treated with LPS exhibited a substantial increase in the production of NO, intracellular ROS, and SEAP compared to the control group (Figure 4). Among the extracts, the leaf extract of Geisha (43.8 ± 2.75%) and the green bean extract of Costarica (63.1 ± 1.19%) exhibited the most effective NO inhibition.

The coffee extracts also showed a considerable inhibition of ROS production when compared to the group treated with LPS (Figure 4B). Among the leaves, Geisha showed significant efficacy, resulting in a 31.5% reduction in ROS levels compared to the LPS control group. In contrast, green beans had no notable differences in the ROS inhibition between cultivars.

Among the leaves, only Geisha showed a substantial inhibition of LPS-induced NF-κB activation (74.1 ± 6.75%) (Figure 4C). In contrast, with the exception of Catuai, all green bean extracts showed a substantially inhibited SEAP level compared to LPS, but no significant variations were seen among cultivars.

Ding et al. reported coffee leaves’ anti-inflammatory activities, including reducing the production of nitric oxide in RAW 264.7 cells stimulated with LPS [48]. Other research also demonstrated that coffee seeds decreased pro-inflammatory cytokines, including IL-6, TNF-α, and IL-1β, in THP-1-derived macrophages [37]. Within the same investigation, it was shown that green bean extracts exhibited a 50% suppression of an SEAP increase at a concentration of 250 µg/mL. Inflammation increases oxidative stress, creating a feedback loop that further stimulates immune responses and activates inflammatory pathways [49]. Oxidative stress regulates the various enzymes that are involved in the generation of ROS, leading to additional damage [50]. Our results emphasize the possible anti-inflammatory advantages of coffee leaf and green bean extracts, indicating that specific cultivars and components may provide improved health benefits. Additional investigations should examine the fundamental mechanisms and environmental factors that affect these bioactive qualities.

Green bean and leaf extracts did not show a significant correlation between the active compounds and antioxidant efficacy. The efficacy of polyphenols and flavonoids in radical scavenging can fluctuate based on factors such as structural attributes, non-covalent interactions, and the existence of competing reactive species. The quantity and arrangement of hydroxyl groups, as observed in morin compared to quercetin-7-D-glucoside, affect the scavenging efficacy [51]. Furthermore, non-covalent interactions such as hydrogen bonding may either impede or augment antioxidant activity [52]. A comprehensive study of polyphenols and flavonoids in coffee leaves and green beans is vital, given the impact of these diverse components. Meanwhile, the TPC of leaf extracts showed a significant association with NF-κB inhibition, with a correlation coefficient of r = −0.79 (Table 2). Additionally, NF-κB activity levels were highly correlated with NO and ROS, and the TFC of leaf extracts exhibited a significant negative correlation with anti-inflammatory efficacies. Multiple studies have demonstrated that flavonoids decrease NF-κB activity in inflammatory conditions [53,54].

The results of our study did not show significant correlations between CGA, caffeine, and mangiferin and the antioxidant and anti-inflammatory efficacy. Although these compounds are reported to have excellent radical scavenging and antioxidant properties, it is crucial to consider that in a mixture, the antioxidant properties of individual ingredients do not consistently demonstrate an additive effect [55]. Ding et al. reported that mono-CQAs in coffee leaves do not contribute to anti-inflammatory efficacy [48]. However, other studies suggest that 5-CQA hinders the activation of NF-κB produced by LPS [56]. Some research suggests that 5-CQA enhances the expression of M1 macrophage indicators caused by LPS/IFN-γ [57], while other research indicates that CGA effectively suppresses the activation of the NLRP3 inflammasome in a mouse model of pneumonia [58]. The presence of conflicting evidence regarding the anti-inflammatory effects indicates an intricate interaction between different CGAs and inflammation. Ding et al. proposed that higher amounts of caffeine hinder the effectiveness of anti-inflammatory properties, while mangiferin plays a role in promoting anti-inflammatory effects [48]. In contrast, other research has demonstrated that caffeine has a notable impact on reducing the production of NO and the activation of NF-κB in Raw264.7 cells by increasing IκB levels [59]. This suggests that caffeine plays a multifaceted function in regulating inflammation, which requires additional exploration. These results suggest that the bioactive compounds in coffee leaves and green beans might have dual roles in inflammation regulation, highlighting the need for additional research to elucidate the precise mechanisms involved. Additionally, the use of a standardized extraction method aimed to minimize the introduction of extraneous variables that might affect the bioactive compounds in the samples; however, this consistency may also be regarded as a limitation. It is well known that coffee undergoes various changes, including in its bioactive compounds and flavor components, during the roasting process. Therefore, further research is needed to investigate the impact of processing methods, such as roasting and drying, on the functionality of coffee across different cultivars.

The analysis of polyphenolic chemicals in coffee leaves and green beans underscores the unexploited potential of coffee leaves as a functional ingredient. Coffee leaves are typically discarded during cultivation to improve the bean yield, but their higher levels of beneficial chemicals relative to green beans indicate their potential as a sustainable resource. This work establishes a basis for subsequent research on the functional and economic uses of coffee leaves in the creation of functional products.

### 3.5. Volatile Compounds

The relative peak area (RPA) of volatile compounds exhibited considerable variations between the leaf and green bean extracts of seven Arabica coffee cultivars, as shown in Table 3. The RPA of fatty acids, fatty acid esters, fatty alcohols, and sterols varied between 45.83 and 57.76% in leaves and between 25.55 and 40.88% in green beans. The Venecia leaf and Obata green bean extract exhibited the highest RPA among the cultivars. Pentadecanoic acid was the most abundant fatty acid, accounting for 16.19 to 21.38% of the RPA in leaves and 6.10 to 15.72% in green beans. A study conducted on Robusta green beans from China revealed that the most abundant fatty acids present were linolenic, palmitic, and oleic acids, along with eicosenoic, myristic, and tricosanoic acids [60]. Furthermore, the research by Martin et al. categorizes Arabica cultivars by their levels of myristic, oleic, and linoleic acids [58]. The linoleic acid methyl ester was exclusively found in green beans (RPA: 1.09% in Caturra to 10.80% in Obata), while fatty alcohols, particularly phytol (RPA: 1.74 to 2.36%), were identified only in leaves. Phytol, an isoprenoid alcohol, is vital in attaching chlorophyll to thylakoid membranes in plant leaves [61]. It is also known for its capacity to reduce lipids, enhance insulin sensitivity, and function as an antimicrobial agent; additionally, it serves as a precursor of tocopherol [62,63]. Carrera F et al. investigated coffee sterols and highlighted high levels of stigma sterol [64], a finding supported by our study’s detection of stigma sterol and Stigmast-5-en-3-ol. Collectively, these results underscore not only the varietal but also the cultivar-specific differences in coffee fatty acids, emphasizing their biological significance.

The volatile compounds of the phenol group found in this study comprise 2-Methoxy-4-vinylphenol, acetoveratrone, α-Tocopherol-β-D-mannoside, β-tocopherol, γ-tocopherol, and quinic acid. Out of these compounds, α-Tocopherol-β-D-mannoside showed the highest RPA in leaves, with values ranging from 1.86 to 3.06%. Conversely, β-tocopherol was primarily detected in green beans, with RPA levels ranging from 0.73 to 1.18%. Tocopherols, also known as vitamin E, are well-known lipophilic antioxidants [65]. They have a vital function in the sensitivity of plants to light and their ability to tolerate temperature stress through their synthesis [66]. While both α- and β-tocopherols have been reported to modulate cell division and proliferation through different mechanisms in various cell lines [67], there remains a lack of comprehensive research. Several studies indicate that α-tocopherol demonstrates superior antioxidant activity in comparison to β-tocopherol [68]. Our findings suggest that the superior antioxidant activity observed in leaf extracts compared to green bean extracts could be attributed to these compounds. Phyran, a heterocyclic molecule consisting of a six-membered ring containing one oxygen atom and five carbon atoms, is now being investigated for its possible anti-inflammatory properties [69,70]. Our analysis found the presence of 4H-Pyran-4-one in both leaf extracts and Catuai green beans. Due to the lack of sufficient research on this molecule in coffee, additional investigations are necessary to clarify its functional activities.

Our investigation discovered that the RPA of caffeine in leaves (ranging from 24.24 to 36.12%) was higher than that in green beans (ranging from 11.03 to 20.36%). Previous research, such as the study using GC-MS and LC-MS for alkaloid analysis [71], highlighted the variability in detected compounds depending on the analytical method employed. Another study highlighted discrepancies between GC-MS and HPLC results, emphasizing the necessity of using alternate methodologies in analytical methods [72]. The results of our study on caffeine levels also show that there is variation due to the way the samples were prepared and the procedures used for analysis. These observations highlight the significance of maintaining methodological consistency and validation in caffeine analysis across various coffee components to provide accurate comparison investigations. Our findings regarding caffeine levels indicate variability attributable to the sample preparation and analytical methods employed.

PCA is a useful tool to reduce dimensions and highlight the most relevant volatile compounds that distinguish samples [73]. Figure 5A presents the scree plot illustrating the eigenvalues of each principal component. The first principal component (PC1) accounts for the majority of the variance in the data, while the second principal component (PC2) elucidates a smaller yet notable portion. After the second component, the eigenvalues decline significantly, suggesting that only the first two principal components are necessary to account for the majority of the variance in the dataset. The significant decline justifies the exclusive utilization of PC1 and PC2 in further analysis, thereby streamlining the data’s dimensionality while preserving essential information. Figure 5B illustrates the hierarchical clustering dendrogram that categorizes the compounds according to their similarity. The distance between clusters reflects the similarity or dissimilarity, with nearby branches signifying that they are more analogous. This analysis may support the findings from the PCA by grouping the volatiles based on hierarchical similarity and figuring out their association with the coffee bean and leaf samples (shown as red circles in Figure 5C). The 3D PCA plot (Figure 5C) depicts the projection of the samples (B~O) and the chemical constituents identified via GC-MS onto the principal components. PC 1 explains 96.2% of the total variance, representing the largest source of variation among the chemical compounds. PC 2 explains 2.9% of the variance, and PC 3 explains 0.8% of the variance. Together, these three components account for 99.9% of the total variance, indicating that most of the differences between the samples are captured by these three principal components. The arrows represent the loading vectors of each sample. The leaves (B to H) and green bean (I to O) extracts showed different directions. Caffeine, showing a high contribution to PC1, was separated from several loading vectors, which indicates its independent characteristics, being less influenced by the samples. Pentadecanoic acid appeared to be uniformly influenced by PC1 and PC2, and an examination of its relationship with the loading vector suggests a stronger association with green beans than with leaves.

The leaf samples (B to H) were primarily associated with compounds like octadecanoic acid, Stigmast-5-en-3-ol, and caffeine; these compounds are linked to the lower portion of the PCA plot. Octadecanoic acid, in particular, was strongly associated with the leaf, especially in Costarica (D), Geisha (E), and Marsellesa (F).

In contrast, the green bean (I to O) samples were positioned toward the upper right and were associated with different compounds, including pentadecanoic acids, stigmasterol, and 2-palmitoylglycerol. These compounds suggest that green beans have a distinct chemical composition that is rich in different types of fatty acids and lipid metabolites.

## 4. Conclusions

This research investigated the bioactive compounds and their antioxidant and anti-inflammatory characteristics in the leaves and green beans of seven Arabica coffee cultivars cultivated in a controlled greenhouse setting. Furthermore, GC-MS and PCA provided insights into the volatile compounds found in the leaves and green beans. The leaf extract of Geisha was an exceptional source of polyphenols and flavonoids, exhibiting considerable antioxidant and anti-inflammatory properties. The Marsellesa cultivar was a potent source of flavonoids from green beans and mangiferin from the leaves, which demonstrate exceptional antioxidant and anti-inflammatory properties.

Our findings support not only variations among cultivars but also the use of coffee leaves as a viable alternative to green beans, which has substantial implications for the coffee industry and health-related applications. Utilizing the biologically active compounds in coffee leaves is expected to produce innovative products that provide economic and health benefits, thus promoting the sustainable development of coffee farming in temperate areas.

## Figures and Tables

**Figure 1 foods-13-03744-f001:**
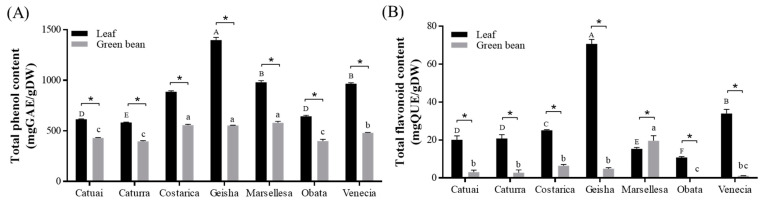
Total phenol (**A**) and flavonoid (**B**) contents in the leaves and green bean extracts of seven cultivars of Arabica coffee cultivated in a greenhouse. Data represent the mean ± SEM (*n* = 3). Different uppercase letters indicate significant differences among leaf groups, while lowercase letters indicate significant differences among green bean groups, as determined by Tukey’s multiple comparison test at *p* < 0.05. * Indicates significant differences between leaf and green bean extracts at *p* < 0.05.

**Figure 2 foods-13-03744-f002:**
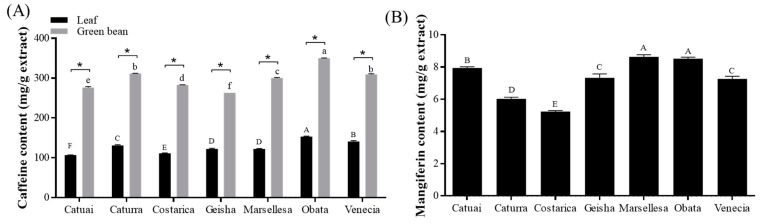
The caffeine (**A**) and mangiferin (**B**) contents in the leaves and green bean extracts of seven cultivars of Arabica coffee cultivated in a greenhouse. Data represent the mean ± SEM. Different uppercase letters indicate significant differences among leaf groups, while lowercase letters indicate significant differences among green bean groups, as determined by Tukey’s multiple comparison test at *p* < 0.05. * Indicates significant differences between leaf and green bean extracts at *p* < 0.05.

**Figure 3 foods-13-03744-f003:**
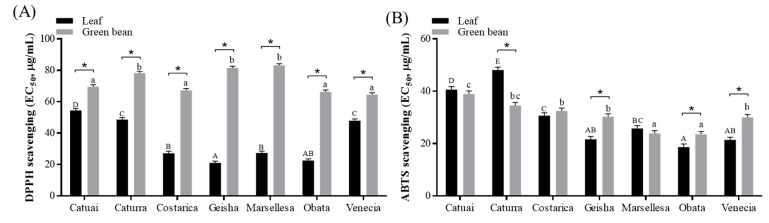
DPPH (**A**) and ABTS (**B**) radical scavenging efficacy of the leaves and green bean extracts of seven cultivars of Arabica coffee cultivated in a greenhouse. Data represent the mean ± SEM (*n* = 3). Different uppercase letters indicate significant differences among leaf groups, while lowercase letters indicate significant differences among green bean groups, as determined by Tukey’s multiple comparison test at *p* < 0.05. * Indicates significant differences between leaf and green bean extracts at *p* < 0.05.

**Figure 4 foods-13-03744-f004:**
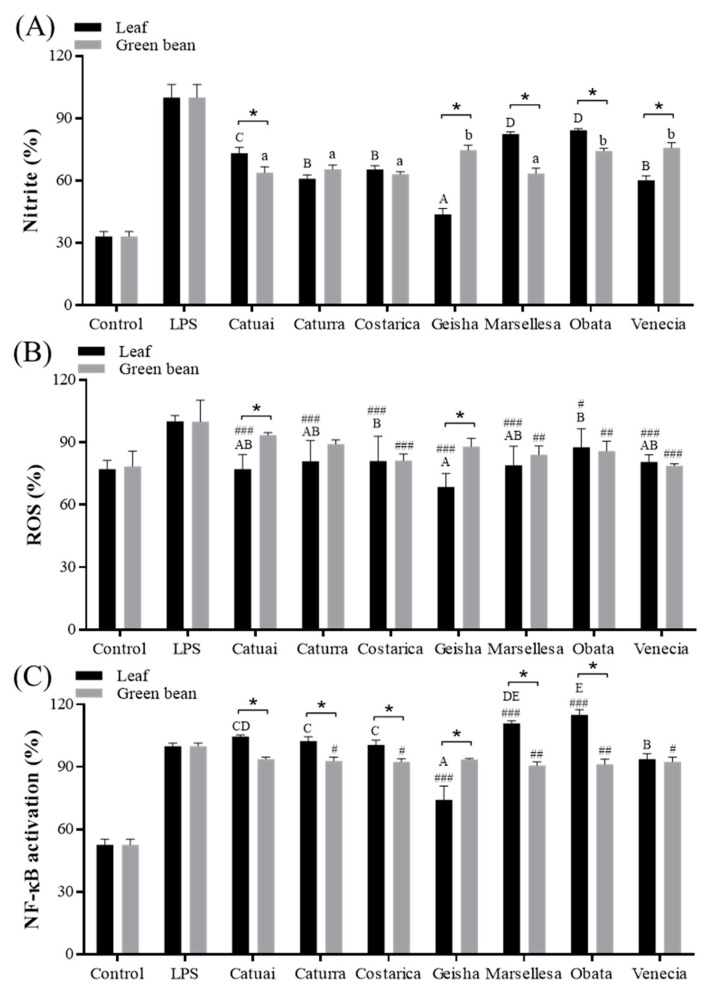
Nitrite (**A**), ROS (**B**), and SEAP (**C**) production-inhibiting effects of the leaves and green bean extracts of seven cultivars of Arabica coffee cultivated in a greenhouse. Inflammatory factors were measured in 400 μg/mL of extracts and LPS (1 μg/mL)-treated Raw-BLUETM cells. The production ratio (%) was calculated based on the LPS-treated control. Data represent the mean ± SEM. Different uppercase letters indicate significant differences among leaf groups, while lowercase letters indicate significant differences among green bean groups, as determined by Tukey’s multiple comparison test at *p* < 0.05. * Indicates significant differences between leaf and green bean extracts at *p* < 0.05. #, ##, and ### indicate significant differences to LPS at *p* < 0.05, 0.01, and 0.001.

**Figure 5 foods-13-03744-f005:**
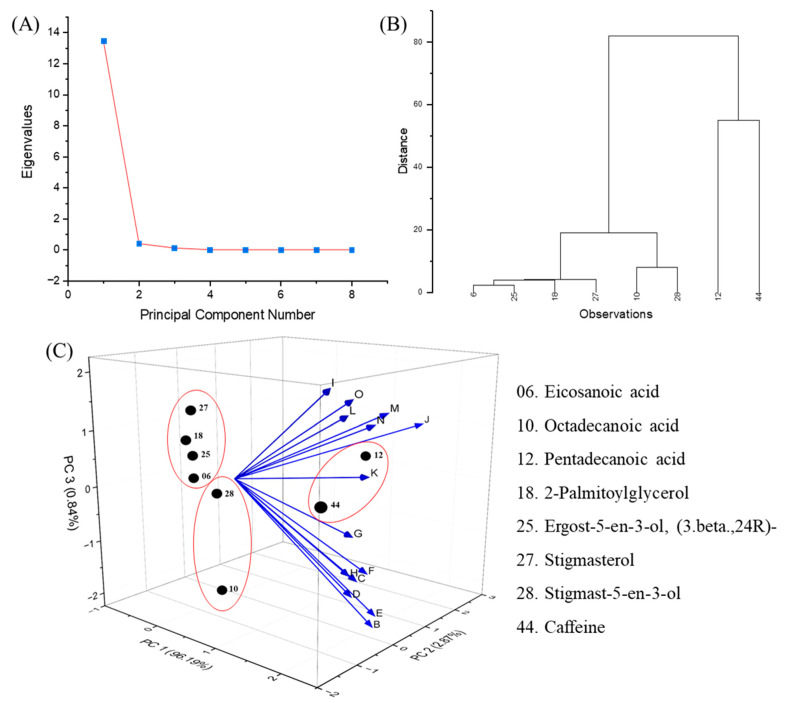
Comprehensive analysis of coffee bean and leaf samples using PCA and hierarchical clustering. The scree plot (**A**) displays the eigenvalues for each principal component derived from the GC-MS data. The dendrogram (**B**) displays the hierarchical clustering of samples based on chemical similarity. Samples with closer chemical profiles cluster together, with the height of the branches reflecting the degree of dissimilarity. The 3D PCA plot (**C**) visualizes the distribution of chemical compounds based on the first three principal components (PC1 = 96.19%, PC2 = 2.87%, PC3 = 0.84%). B, Catuai leaf; C, Caturra leaf; D, Costarica leaf; E, Geisha leaf; F, Marsellesa leaf; G, Obata leaf; H, Venecia leaf; I, Catuai green bean; J, Caturra green bean; K, Costarica green bean; L, Geisha green bean; N, Marsellesa green bean; M, Obata green bean; O, Venecia green bean.

**Table 1 foods-13-03744-t001:** The chlorogenic acid contents in the leaves and green bean extracts of seven cultivars of Arabica coffee cultivated in a greenhouse. Data represent the mean ± SEM (*n* = 3). Different uppercase letters indicate significant differences among leaf groups, while lowercase letters indicate significant differences among green bean groups, as determined by Tukey’s multiple comparison test at *p* < 0.05.

Cultivar	Part	3-CQA (mg/g)	4-CQA (mg/g)	5-CQA (mg/g)	4-FQA (mg/g)	5-FQA (mg/g)	3,5-DiCQA (mg/g)	4,5-DiCQA (mg/g)	Total (mg/g)
Catuai	Leaf	0.8 ± 0.00 ^A^	1.4 ± 0.03 ^A^	323.1 ± 1.19 ^AB^	0.1 ± 0.02 ^CD^	0.2 ± 0.00 ^AB^	1.3 ± 0.02 ^D^	0.5 ± 0.12 ^B^	327.4 ± 1.34 ^AB^
	Green bean	5.8 ± 0.05 ^e^	8.2 ± 0.06 ^c^	64.3 ± 3.18	0.5 ± 0.02 ^c^	4.4 ± 0.05 ^d^	6.4 ± 0.05 ^c^	2.7 ± 0.03 ^bc^	92.5 ± 2.98 ^ab^
Caturra	Leaf	0.7 ± 0.01 ^B^	1.2 ± 0.01 ^B^	321.0 ± 5.54 ^AB^	0.1 ± 0.02 ^CD^	0.2 ± 0.01 ^AB^	1.6 ± 0.03 ^C^	0.5 ± 0.07 ^B^	325.3 ± 5.64 ^ABC^
	Green bean	4.7 ± 0.00 ^g^	7.0 ± 0.01 ^e^	61.1 ± 3.36	0.5 ± 0.01 ^c^	5.2 ± 0.02 ^b^	8.1 ± 0.01 ^a^	2.6 ± 0.01 ^bc^	89.6 ± 3.40 ^ab^
Costarica	Leaf	0.7 ± 0.01 ^AB^	1.2 ± 0.02 ^BC^	267.6 ± 3.16 ^D^	0.3 ± 0.00 ^A^	0.3 ± 0.00 ^AB^	1.1 ± 0.02 ^E^	0.6 ± 0.01 ^AB^	271.5 ± 3.21 ^E^
	Green bean	6.7 ± 0.01 ^b^	9.1 ± 0.02 ^b^	58.9 ± 2.14	0.6 ± 0.01 ^b^	4.2 ± 0.02 ^e^	5.9 ± 0.01 ^d^	1.8 ± 0.00 ^d^	87.7 ± 2.14 ^b^
Geisha	Leaf	0.7 ± 0.02 ^AB^	1.2 ± 0.03 ^B^	286.8 ± 6.49 ^C^	0.2 ± 0.01 ^BC^	0.2 ± 0.01 ^B^	1.8 ± 0.04 ^B^	0.6 ± 0.01 ^AB^	291.4 ± 6.59 ^D^
	Green bean	5.9 ± 0.02 ^d^	8.2 ± 0.01 ^c^	61.9 ± 3.42	0.4 ± 0.00 ^d^	3.5 ± 0.01 ^f^	6.0 ± 0.01 ^d^	2.5 ± 0.01 ^c^	88.5 ± 3.43 ^ab^
Marsellesa	Leaf	0.5 ± 0.01 ^C^	1.1 ± 0.02 ^C^	324.3 ± 4.97 ^A^	0.1 ± 0.00 ^D^	0.3 ± 0.00 ^A^	1.9 ± 0.03 ^A^	0.7 ± 0.01 ^A^	328.9 ± 5.03 ^B^
	Green bean	5.2 ± 0.01 ^f^	7.3 ± 0.02 ^d^	54.5 ± 1.87	0.6 ± 0.00 ^a^	5.4 ± 0.03 ^a^	6.4 ± 0.03 ^c^	2.8 ± 0.01 ^b^	82.5 ± 1.76 ^b^
Obata	Leaf	0.7 ± 0.0 ^AB^	1.2 ± 0.01 ^B^	320.4 ± 3.89 ^AB^	0.1 ± 0.00 ^BC^	0.3 ± 0.00 ^AB^	1.7 ± 0.02 ^BC^	0.7 ± 0.01 ^A^	325.1 ± 3.95 ^ABC^
	Green bean	6.2 ± 0.01 ^c^	8.3 ± 0.01 ^c^	60.8 ± 3.60	0.6 ± 0.00 ^b^	4.5 ± 0.01 ^c^	7.4 ± 0.02 ^b^	3.5 ± 0.01 ^a^	91.6 ± 3.59 ^ab^
Venecia	Leaf	0.8 ± 0.02 ^A^	1.3 ± 0.03 ^AB^	310.3 ± 7.11 ^B^	0.2 ± 0.01 ^B^	0.3 ± 0.01 ^AB^	1.6 ± 0.04 ^BC^	0.7 ± 0.02 ^A^	315.2 ± 7.22 ^C^
	Green bean	6.9 ± 0.03 ^a^	9.5 ± 0.05 ^a^	70.9 ± 3.49	0.6 ± 0.00 ^b^	4.4 ± 0.02 ^d^	4.1 ± 0.02 ^e^	2.0 ± 0.01 ^d^	98.5 ± 1.68 ^a^

**Table 2 foods-13-03744-t002:** Correlation coefficients for the relationships between the total phenol content (TPC), total flavonoid content (TFC), total CGAs, caffeine, and mangiferin and the antioxidative and anti-inflammatory efficacy of Arabica coffee cultivated in a greenhouse. * Indicates significance at *p* < 0.05.

Part	Function	TPC (mg GAEs/g)	TFC (mg QUEs/g)	Total CGA (mg/g)	Caffeine (mg/g)	Mangiferin (mg/g)	DPPH	ABTS	NO	ROS
Leaf	DPPH (EC50, μg/mL)	−0.54	−0.26	0.44	−0.19	−0.14	-	-	-	-
	ABTS (EC50, μg/mL)	−0.58	−0.28	0.21	−0.47	−0.44	0.67	-	-	-
	NO (%)	−0.61	−0.89 *	0.55	0.20	0.53	−0.04	−0.06	-	-
	ROS (%)	−0.74	−0.85 *	0.32	0.56	0.05	0.07	−0.02	0.72	-
	NF-kB (%)	−0.79 *	−0.98 *	0.55	0.22	0.31	0.11	0.17	0.95 *	0.84 *
Green bean	DPPH (EC50, μg/mL)	0.40	0.65	−0.73	−0.36	-	-	-	-	-
	ABTS (EC50, μg/mL)	−0.28	−0.37	0.30	−0.56	-	−0.14	-	-	-
	NO (%)	−0.16	−0.53	0.59	0.29	-	−0.25	−0.38	-	-
	ROS (%)	−0.44	−0.14	−0.15	−0.24	-	0.34	0.51	−0.28	-
	NF-κB (%)	−0.19	−0.53	0.40	−0.65	-	−0.07	0.87 *	0.04	0.49

**Table 3 foods-13-03744-t003:** Relative area of volatile compounds identified in the leaves and green bean extracts of seven cultivars of Arabica coffee cultivated in a greenhouse. N.D. means not detected.

Group	No.	Chemical Name	Peak Area (%)													
			Leaf							Seed						
			Catuai	Caturra	Costarica	Geisha	Marsellesa	Obata	Venecia	Catuai	Caturra	Costarica	Geisha	Marsellesa	Obata	Venecia
Fatty acid	1	2,6,10,14,18-Pentamethyl-2,6,10,14,18-eicosapentaene	4.91	2.82	3.74	2.04	0.71	1.69	1.60	N.D.	N.D.	N.D.	N.D.	N.D.	N.D.	N.D.
	2	9-Octadecenamide	3.84	1.64	0.68	2.57	0.57	0.87	0.46	N.D.	N.D.	N.D.	N.D.	N.D.	N.D.	N.D.
	3	Docosanoic acid	N.D.	N.D.	N.D.	N.D.	N.D.	N.D.	N.D.	0.53	0.62	0.75	N.D.	0.97	0.56	0.65
	4	Dodecanoic acid	N.D.	N.D.	N.D.	N.D.	N.D.	N.D.	N.D.	0.57	0.57	0.89	N.D.	0.25	0.18	N.D.
	5	Eicosanoic acid	2.31	2.18	2.18	3.29	2.31	1.80	1.74	1.09	1.63	2.22	1.07	1.98	1.46	1.69
	6	Heptadecanoic acid	0.88	1.38	0.93	N.D.	0.78	0.67	0.65	N.D.	N.D.	N.D.	N.D.	N.D.	N.D.	N.D.
	7	Methyl oleate	N.D.	N.D.	N.D.	N.D.	N.D.	N.D.	N.D.	0.60	N.D.	N.D.	0.49	N.D.	0.38	0.70
	8	Nonadecanoic acid	0.67	N.D.	0.56	0.58	0.39	N.D.	0.41	N.D.	N.D.	N.D.	N.D.	N.D.	N.D.	N.D.
	9	Octadecanoic acid	10.02	9.61	9.66	9.64	9.94	7.76	7.61	2.66	4.64	5.66	3.87	4.63	4.13	3.75
	10	Oleic acid	N.D.	N.D.	N.D.	N.D.	N.D.	N.D.	N.D.	3.32	2.20	0.51	6.46	5.48	2.13	0.66
	11	Pentadecanoic acid	18.30	19.34	17.52	19.30	21.84	17.92	14.43	12.08	30.90	18.12	14.97	20.47	16.78	15.51
	12	Tetradecanoic acid	0.72	0.42	0.62	0.51	0.53	0.49	0.43	N.D.	N.D.	N.D.	N.D.	0.27	0.18	0.39
Fatty acid ester	13	Linoleic acid methyl ester	N.D.	N.D.	N.D.	N.D.	N.D.	N.D.	N.D.	10.87	1.09	1.25	10.83	13.74	14.65	2.90
	14	Linoleic acid ethyl ester	N.D.	0.50	N.D.	1.03	N.D.	0.68	0.52	N.D.	N.D.	N.D.	N.D.	N.D.	N.D.	N.D.
	15	2-Linoleoylglycerol	N.D.	N.D.	N.D.	N.D.	N.D.	N.D.	N.D.	1.33	1.23	1.32	N.D.	2.18	2.44	N.D.
	16	Linolenelaidic acid methyl ester	0.48	0.33	N.D.	0.43	0.35	0.36	17.74	N.D.	N.D.	N.D.	N.D.	N.D.	N.D.	N.D.
	17	2-Palmitoylglycerol	1.40	1.02	1.00	1.00	0.96	0.85	0.76	0.97	0.96	0.88	1.24	2.03	1.90	3.03
	18	Palmitic acid methyl ester	0.48	0.32	0.29	N.D.	0.39	0.33	0.35	2.60	1.00	1.20	2.31	1.58	1.60	2.57
	19	Hexanedioic acid, bis(2-ethylhexyl) ester	0.89	0.39	N.D.	N.D.	N.D.	N.D.	N.D.	3.20	2.50	2.70	1.75	1.58	1.17	1.52
	20	2-Stearoylglycerol	3.30	0.96	N.D.	1.63	N.D.	0.91	0.52	N.D.	3.71	N.D.	N.D.	N.D.	N.D.	N.D.
Fatty alchol	21	1-Eicosanol	N.D.	0.35	0.30	N.D.	0.33	N.D.	N.D.	N.D.	N.D.	N.D.	N.D.	N.D.	N.D.	N.D.
	22	Linolenyl alcohol	N.D.	0.35	0.30	N.D.	0.33	N.D.	N.D.	N.D.	N.D.	N.D.	N.D.	N.D.	N.D.	N.D.
	23	Phytol	2.67	2.17	2.18	1.74	2.31	2.31	2.01	N.D.	N.D.	N.D.	N.D.	N.D.	N.D.	N.D.
Sterol	24	Ergost-5-en-3-ol, (3.beta.,24R)-	2.44	1.65	2.11	1.96	1.70	2.10	1.68	2.17	2.34	2.43	2.09	2.13	1.46	1.83
	25	Fucosterol	N.D.	0.50	0.47	N.D.	0.39	0.41	N.D.	N.D.	N.D.	N.D.	N.D.	N.D.	N.D.	N.D.
	26	Stigmasterol	1.74	1.08	1.24	1.10	1.21	1.31	1.14	3.38	2.54	2.79	3.34	3.26	2.54	3.07
	27	Stigmast-5-en-3-ol	7.02	7.45	7.74	6.34	6.75	6.33	5.71	5.25	6.49	7.02	5.71	4.31	3.90	5.56
Phenol	28	2-Methoxy-4-vinylphenol	1.01	N.D.	1.01	0.43	0.77	0.51	0.42	N.D.	N.D.	N.D.	N.D.	N.D.	N.D.	N.D.
	29	Acetoveratrone	N.D.	0.46	0.35	N.D.	N.D.	0.35	N.D.	N.D.	N.D.	N.D.	N.D.	N.D.	N.D.	N.D.
	30	.alpha.-Tocopherol-.beta.-D-mannoside	2.99	2.37	3.10	2.17	1.99	1.90	2.74	0.50	N.D.	N.D.	0.53	0.40	0.47	0.60
	31	.beta.-Tocopherol	N.D.	N.D.	N.D.	N.D.	N.D.	N.D.	N.D.	1.45	1.59	1.84	1.45	1.46	1.17	1.83
	32	.gamma.-Tocopherol	N.D.	N.D.	1.01	0.49	N.D.	N.D.	0.41	N.D.	N.D.	N.D.	N.D.	N.D.	N.D.	N.D.
	33	Quinic acid	N.D.	N.D.	N.D.	7.87	N.D.	4.55	1.09	N.D.	N.D.	N.D.	N.D.	N.D.	N.D.	N.D.
Triterpenoid	34	24-Methylenecycloartanol	0.48	0.46	0.54	N.D.	0.43	0.47	0.43	1.03	1.51	1.78	1.02	0.51	0.80	0.66
	35	Cycloartenol	1.45	1.08	1.86	0.54	0.78	1.11	0.95	N.D.	0.91	1.10	N.D.	N.D.	N.D.	N.D.
Benzofuran	36	Coumaran	N.D.	N.D.	N.D.	N.D.	N.D.	N.D.	N.D.	5.32	N.D.	N.D.	6.00	0.87	11.83	16.95
	37	Loliolide	0.52	0.64	0.74	0.59	0.78	0.75	0.57	N.D.	N.D.	N.D.	N.D.	N.D.	N.D.	N.D.
Carboxylic acid	38	Benzenepropanoic acid, beta.-amino-4-methoxy-	N.D.	1.56	N.D.	1.18	N.D.	0.89	N.D.	5.62	4.09	5.02	3.96	3.06	2.49	3.23
Benzenediol	39	Pyrocatechol	N.D.	N.D.	N.D.	1.32	1.49	1.00	1.03	N.D.	N.D.	N.D.	N.D.	0.96	0.94	1.28
Diterpene	40	Neophytadiene	2.38	2.03	2.11	2.16	1.88	2.46	2.01	N.D.	N.D.	N.D.	N.D.	N.D.	N.D.	N.D.
Alchol	41	Solanesol	0.48	N.D.	1.74	N.D.	N.D.	N.D.	0.87	N.D.	N.D.	N.D.	N.D.	N.D.	N.D.	N.D.
Aldehyde	42	5-Hydroxymethylfurfural	N.D.	N.D.	N.D.	0.69	1.63	1.27	0.71	1.45	7.81	12.01	N.D.	N.D.	N.D.	N.D.
Alkaloid	43	Caffeine	27.40	36.96	35.10	28.45	36.90	36.75	30.24	33.16	21.67	30.51	32.89	27.88	26.86	31.61
Pyranone	44	4H-Pyran-4-one	1.24	N.D.	0.93	0.97	1.53	1.18	0.76	0.85	N.D.	N.D.	N.D.	N.D.	N.D.	N.D.

## Data Availability

The original contributions presented in the study are included in the article, further inquiries can be directed to the corresponding author.

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
