# Peer review of "Comparative Analysis of Phytochemical and Functional Profiles of Arabica Coffee Leaves and Green Beans Across Different Cultivars"

_foods, 2024, doi:10.3390/foods13233744_

Round 1
Reviewer 1 Report
Comments and Suggestions for Authors
The feasibility of the described studies should be critically assessed in the context of availability and profitability. Coffee leaves are less available and more difficult to obtain on an industrial scale than seeds, which may limit their use in commercial production. Obtaining them may weaken the bushes and beans. In addition, bioactive components in leaves are present in smaller quantities than in fruits, which reduces their potential effectiveness and would require significant investments in extraction and concentration technology.
In the summary, the authors state that the determinations were made using colorimetric methods, which are rarely used today. The description of the methods describes spectrophotometric methods. Colorimetric methods are based on the assessment of the intensity of the color of the solution, which is created after the chemical reaction of the sample with the appropriate coloring reagent. The result is assessed visually (subjective assessment of color) or using a colorimeter. Isn't that what the authors had in mind?
The authors used a temperature of 50 degrees C to prepare the extract. Were they not afraid that some polyphenolic compounds would decompose at this temperature? Such temperatures are not usually used to determine these compounds.
How does the method used by the authors differ from the traditional Folin-Ciocalteu method? The only difference I see is the wavelength. Please also explain whether it is possible to detect most polyphenols at this wavelength? Usually, a wavelength of 765 nm is used.
Total polyphenols and total flavonoids results should be expressed per dry matter rather than per gram of extract.
Figure 1 shows the differences in the content of polyphenols and flavonoids in leaves and green coffee beans. It is not really comparable because the results are calculated per gram of extract, not dry matter content. It is also known that the content of these compounds will be different. So is there any point in determining the significance of differences between these extracts?
Some of the items on which the literature review is based could be newer. There are many reports on this topic.
Author Response
Reviewer 1
[Comment] The feasibility of the described studies should be critically assessed in the context of availability and profitability. Coffee leaves are less available and more difficult to obtain on an industrial scale than seeds, which may limit their use in commercial production. Obtaining them may weaken the bushes and beans. In addition, bioactive components in leaves are present in smaller quantities than in fruits, which reduces their potential effectiveness and would require significant investments in extraction and concentration technology.
[Responses] We appreciate the reviewer's insightful comments on the feasibility of using coffee leaves in commercial production. We have addressed these concerns as follows:
Traditionally regarded as by-products, coffee leaves are recognized for their potential as sustainable and functional ingredients, rich in bioactive compounds. The standard agricultural practices of leaf removal and pruning, necessary to optimize coffee bean yield, present an opportunity to repurpose these leaves into value-added products like functional beverages rather than allowing them to go to waste. Our study further demonstrated that coffee leaves contain higher levels of TPC, TFC, CGA, and mangiferin than beans, highlighting their potential value. While we acknowledge that scaling up coffee leaf production poses challenges, we agree that further research on this process's economic and technological aspects is essential to realize its commercial potential.
These revisions have been incorporated into the manuscript as follows: p.2, lines 59-62 and 64-65. Thank you for your valuable suggestions.
[Comment] In the summary, the authors state that the determinations were made using colorimetric methods, which are rarely used today. The description of the methods describes spectrophotometric methods. Colorimetric methods are based on the assessment of the intensity of the color of the solution, which is created after the chemical reaction of the sample with the appropriate coloring reagent. The result is assessed visually (subjective assessment of color) or using a colorimeter. Isn't that what the authors had in mind?
[Responses] Thank you for your detailed observation regarding the description of the colorimetric methods in our study. We agree that the term "colorimetric" might have caused some confusion. To clarify, the colorimetric method used in our study employed a spectrophotometer to measure the absorbance of the solution, which indirectly infers the concentration through changes in color intensity. This approach aligns with modern spectrophotometric techniques rather than traditional visual or colorimeter-based methods.
We have revised the text to clarify this distinction, as noted in p.1, line 18. Thank you for bringing this to our attention.
[Comment] The authors used a temperature of 50 degrees C to prepare the extract. Were they not afraid that some polyphenolic compounds would decompose at this temperature? Such temperatures are not usually used to determine these compounds.
[Responses] Thank you for your concern regarding the extraction temperature. We ensured minimal exposure to high temperatures during the process to preserve polyphenolic compounds. Studies show that temperatures up to 60°C do not affect polyphenol or tannin yields compared to freeze-drying [1], and no significant changes in antioxidant activity are observed at temperatures as high as 120–150°C [2]. Additionally, the optimal extraction of polyphenols is often achieved at 60–80°C [3, 4]. Based on this evidence, we selected 50°C as a suitable temperature to balance effective extraction and compound stability. Thank you for raising this important point.
[Comment] How does the method used by the authors differ from the traditional Folin-Ciocalteu method? The only difference I see is the wavelength. Please also explain whether it is possible to detect most polyphenols at this wavelength? Usually, a wavelength of 765 nm is used.
[Responses] Thank you for your question regarding the wavelength used in our study compared to the traditional Folin-Ciocalteu method. While 765 nm is commonly used for optimal polyphenol detection [5], we measured absorbance at 720 nm based on the detection capabilities of our equipment and prior studies supporting its use for this assay [6]. The assay's chromogen also demonstrates increasing absorbance within the 550–750 nm range (F9252, Sigma-Aldrich), making 720 nm a valid choice for detecting polyphenolic compounds. We appreciate the opportunity to clarify this aspect of our methodology.
[Comment] Total polyphenols and total flavonoids results should be expressed per dry matter rather than per gram of extract. Figure 1 shows the differences in the content of polyphenols and flavonoids in leaves and green coffee beans. It is not comparable because the results are calculated per gram of extract, not dry matter content. It is also known that the content of these compounds will be different. So is there any point in determining the significance of differences between these extracts?
[Responses] Thank you for your valuable suggestion regarding expressing total polyphenols (TPC) and total flavonoids (TFC) per dry matter rather than per gram of extract. We agree that this provides a more accurate basis for comparison. Accordingly, we have recalculated TPC and TFC values based on the dry matter weight of the plant material before extraction and drying, adjusting the statistical significance where necessary. The manuscript has incorporated these updates: p3 (lines 108 and 112-113), p5 (lines 209-212, 218-221), and Figure 1.
This adjustment has also been reflected in the correlation analysis and Table 2, with the results updated as noted: p10 (lines 361-362 and 369-372) and Table 2. We appreciate your insightful recommendation and believe this improves the rigor and clarity of our analysis.
[Comment] Some of the items on which the literature review is based could be newer. There are many reports on this topic.
[Responses] Thank you for highlighting the need to include more recent literature. We agree that incorporating the latest findings strengthens the relevance and rigor of our discussion. Accordingly, we have updated several references to reflect newer studies while retaining foundational sources essential to the topic. The updated references include 4, 18, 19, 20, 30, 33, and 34. We appreciate your suggestion and believe it enhances the quality of our manuscript.
References
[1] J.A. Larrauri, P. Rupérez, F. Saura-Calixto, Effect of drying temperature on the stability of polyphenols and antioxidant activity of red grape pomace peels, Journal of agricultural and food chemistry 45(4) (1997) 1390-1393.
[2] C.F. Ross, J. Hoye, Clifford, V.C. Fernandez‐Plotka, Influence of heating on the polyphenolic content and antioxidant activity of grape seed flour, Journal of food science 76(6) (2011) C884-C890.
[3] A. Antony, M. Farid, Effect of temperatures on polyphenols during extraction, Applied Sciences 12(4) (2022) 2107.
[4] M. Dent, V. Dragović-Uzelac, M. Penić, T. Bosiljkov, B. Levaj, The effect of extraction solvents, temperature and time on the composition and mass fraction of polyphenols in Dalmatian wild sage (Salvia officinalis L.) extracts, Food technology and biotechnology 51(1) (2013) 84-91.
[5] O. Folin, W. Denis, On phosphotungstic-phosphomolybdic compounds as color reagents, Journal of biological chemistry 12(2) (1912) 239-243.
[6] S.S. Kim, K.J. Park, S.H. Yun, Y.H. Choi, Bioactive compounds and antioxidant capacity of domestic citrus cultivar ‘Haryejosaeng’, Food Science and Preservation 26(6) (2019) 681-689.
Reviewer 2 Report
Comments and Suggestions for Authors
In the manuscript “Comparative Analysis for Phytochemical and Functional Profiles of Arabica Coffee Leaves and Green Beans Across Different Cultivars”, the comparisons of Arabica Coffee Leaves and Green Beans was demonstrated using chromatography and mass spectrometry. This study has implications for the development of coffee as a functional food product. But, this form of manuscript need to be revised. The comments and questions are as follow:
1. Although this study did relevance analysis, it did insufficiently in-depth in analysis of the correlation between the content of bioactive substances and their functional activities.
2. Line 89-92, Is this the best extraction method for the active substances in coffee beans and leaves? Why is the same treatment used for coffee beans and leaves? The study did not explore the effects of different coffee leaf processing methods (e.g., drying, extraction) on their functional properties.
3. The authors could consider adding sensory evaluations of coffee leaves and green beans, such as taste and aroma, to more fully assess their potential as functional foods.
Author Response
Reviewer 2
[Comment] Although this study did relevance analysis, it did insufficiently in-depth in analysis of the correlation between the content of bioactive substances and their functional activities.
[Responses] Thank you for your valuable feedback regarding the correlation analysis between bioactive substance content and functional activities. We acknowledge that our initial discussion lacked depth in this area. To address this, we have included a more detailed analysis and discussion of these relationships in the revised manuscript: p.10 (lines 362-368 and 376-379). We appreciate your suggestion, which has helped us enhance the scientific rigor of our work.
[Comment] Line 89-92; is this the best extraction method for the active substances in coffee beans and leaves? Why is the same treatment used for coffee beans and leaves? The study did not explore the effects of different coffee leaf processing methods (e.g., drying, extraction) on their functional properties.
[Responses] Thank you for your insightful question regarding the use of the same extraction method for coffee beans and leaves. Our objective was to evaluate varietal differences and the potential of coffee leaves as functional materials. To ensure comparability, we standardized the extraction process using 80% methanol, a widely used solvent for polyphenol extraction [7], under uniform conditions for both beans and leaves. While we acknowledge the importance of exploring different processing methods for coffee leaves, we deliberately minimized variables in this study to focus on intrinsic differences among cultivars grown under consistent conditions. We appreciate your suggestion and plan to investigate the effects of various processing techniques, such as drying and roasting, in future studies.
[Comment] The authors could consider adding sensory evaluations of coffee leaves and green beans, such as taste and aroma, to more fully assess their potential as functional foods.
[Responses] Thank you for your valuable suggestion to include sensory evaluations, such as taste and aroma, to better assess the functional potential of coffee leaves and green beans. We agree that these attributes are essential, particularly as roasting enhances flavor and aroma compounds, which may undergo transformations as indicated by our GC-MS analysis [8]. As part of our ongoing research, we plan to investigate how different roasting conditions influence flavor profiles and aroma compounds across varieties. This will provide a more comprehensive understanding of the sensory attributes and potential applications of coffee leaves as functional ingredients.
[7] M.N. Safdar, T. Kausar, S. Jabbar, A. Mumtaz, K. Ahad, A.A. Saddozai, Extraction and quantification of polyphenols from kinnow (Citrus reticulate L.) peel using ultrasound and maceration techniques, J Food Drug Anal 25(3) (2017) 488-500.
[8] S. Schenker, C. Heinemann, M. Huber, R. Pompizzi, R. Perren, R. Escher, Impact of roasting conditions on the formation of aroma compounds in coffee beans, Journal of food science 67(1) (2002) 60-66.
Reviewer 3 Report
Comments and Suggestions for Authors
The manuscript described the phytochemical composition of leaves and green beans from seven Arabica coffee cultivars, including total phenolic and flavonoid contents, caffeine, chlorogenic acids (CGAs) and mangiferin levels, and volatile compounds. Their functional properties of antioxidant and anti-inflammatory effects were also analyzed. Although the research data and analysis are sufficient and may draw interest of related researchers, there are some concerns should be revised.
1. use appropriate keywords but not abbreviations such as DPPH, NF-kB, NO.
2. It’s worth noting that the components of coffee beans, not only the phenolic and flavonoid contents, but also the flavor components are changed dramatically. And it’s may also the leaves. Thus, it’s much more important to study the chemical components and bioactivities change during coffee beans or leaves process, but not comparing the difference between leaves and beans. For this study, I recommend to supplement discussion on these.
Author Response
Reviewer 3
[Comment] Use appropriate keywords but not abbreviations such as DPPH, NF-kB, NO.
[Responses] Thank you for your observation regarding using abbreviations in keywords. We have revised the keywords to use full terms instead of abbreviations, enhancing clarity and searchability. Additionally, in the manuscript, each abbreviation is introduced with its full name before consistent use of the acronym, as updated in p.1 (lines 19-24), p.2 (lines 80-83), and p.3 (lines 105, 110). We appreciate your suggestion, which has improved the manuscript's readability and accessibility.
[Comment] It is worth noting that the components of coffee beans, not only the phenolic and flavonoid contents but also the flavor components, have changed dramatically. And dismay, also the leaves. Thus, it's much more essential to study the chemical components and bioactivities change during the coffee bean or leaf process, but not compare the difference between leaves and beans. For this study, I recommend supplementing the discussion on these.
[Responses] Thank you for your insightful comments on the importance of studying the changes in chemical components and bioactivities during coffee bean and leaf processing. While we agree that processing significantly alters phenolic, flavonoid, and flavor components, our study primarily focused on examining varietal differences in bioactive potential under identical growing conditions rather than comparing leaves and beans alone. Additionally, since unroasted beans lack fully developed flavor profiles, flavor analysis was not our focus in this study.
We have clarified these objectives in the discussion: p.10 (lines 394-401). Furthermore, we are planning follow-up studies to investigate how roasting, drying, and other processing methods impact high-performing cultivars' bioactivity and sensory attributes. This will allow us to fully explore the functional potential of coffee leaves and beans across various processing conditions. Thank you for your valuable suggestion.
Reviewer 4 Report
Comments and Suggestions for Authors
This research presents a comparative analysis of the phytochemical profiles of leaves and coffee beans from different cultivars. However, it should highlight more what the purpose of analyzing the leaves is, detailing that importance in the manuscript, since it also shows that there are significant differences between leaves and beans. That is, where you want this knowledge to be applied.
In section 2.7. Statistical analyses, it does not mention whether your data have a normal distribution in order to then perform an ANOVA. Otherwise, it will be another statistical test, non-parametric type.
In line 197; the symbol “±” in some data is separated as in line 198 and in others it is next to the data as in line 197. Homogenize throughout the manuscript.
In figure 1. It is suggested to use a single symbol “*” and not three “***”, review the symbology in the figures.
In the figures, the letters “A, B, C and D” are in uppercase in some and in lowercase in others, homogenize.
Line 306; “Costarica” is correct, not “Costa Rica”
Line 380, it is suggested to homogenize the text with the symbol “%”, that is, the correct form is “25.55 and 40.88%”
Review, modify the scientific name in italics.
Author Response
Reviewer 4
[Comment] This research presents a comparative analysis of the phytochemical profiles of leaves and coffee beans from different cultivars. However, it should highlight more what the purpose of analyzing the leaves is, detailing that importance in the manuscript since it also shows that there are significant differences between leaves and beans. That is, where you want this knowledge to be applied.
[Responses] We sincerely appreciate your comprehensive review and insightful feedback on our manuscript. Your recommendation to highlight the objective of examining coffee leaves and to expand on the possible uses of our findings has been crucial in sharpening the focus and significance of our research. In response to your remark, we have amended the manuscript to convey the importance of investigating coffee leaves concerning sustainability and resource management. These revisions have been incorporated into the manuscript as follows: p.2 (lines 59-62 and 64-65).
This modification emphasizes the significance of our research in investigating coffee leaves as an untapped resource, showcasing their potential to enhance sustainable practices and facilitate the creation of innovative commercial goods. We sincerely appreciate your astute observations.
[Comment] In section 2.7. Statistical analyses, it does not mention whether your data have a normal distribution in order to then perform an ANOVA. Otherwise, it will be another statistical test, non-parametric type.
[Responses] We sincerely appreciate your thoughtful and constructive feedback on our manuscript. Your comments regarding the statistical analysis and the need to clarify our approach's assumptions have been invaluable. To address your concerns, we would like to provide additional clarification. Our study involved groups with equal sample sizes, relatively small datasets, and ordinal or ranked data. Non-parametric ANOVA tests were employed to ensure the robustness of our findings. Furthermore, Bartlett's Test and Brown-Forsythe Test were utilized to confirm the assumption of homoscedasticity (equal variance), and these analyses supported the reliability of our statistical results.
Following your suggestion, we have revised the corresponding section in the manuscript as follows: p.5 (lines 192-196).
We are truly grateful for your detailed review.
[Comment] In line 197; the symbol "±" in some data is separated as in line 198 and in others it is next to the data as in line 197. Homogenize throughout the manuscript.
[Responses] We sincerely thank you for your meticulous review and for pointing out the inconsistency in formatting our manuscript's "±" symbol. We sincerely appreciate your attention to detail, which has helped us refine the presentation of our data. In response to your comment, we have carefully reviewed and homogenized the formatting of the "±" symbol throughout the manuscript to ensure consistency. This correction has been implemented in all relevant sections: p.6 (line 252), p.7 (lines 291-294), p.8 (lines 300 and 336).
[Comment] In figure 1. It is suggested to use a single symbol "*" and not three "***," review the symbology in the figures.
[Responses] Thank you for your valuable feedback regarding the symbology used in Figure 1 and throughout the manuscript. We understand the importance of maintaining clarity and consistency when presenting statistical significance. In response to your suggestion, we have revised all figures to use a single symbol, "*," to indicate statistical significance. Additionally, we have included the explanation "p < 0.05" in the figure legends to denote the threshold for significance. These changes have been applied uniformly across all relevant figures to ensure consistency and readability: p.5-6 Figure 1, p.7 Figure 2, p.8 Figure 3, p.9-10 Figure 4.
We greatly appreciate your thoughtful suggestion.
[Comment] In the figures, the letters "A, B, C and D" are in uppercase in some and in lowercase in others, homogenize.
[Responses] We are grateful for your careful review and highlighting the variation in using uppercase and lowercase letters in the figures. This was an intentional design choice, where uppercase letters ("A, B, C, D") were used to indicate statistical differences among leaf groups, and lowercase letters ("a, b, c, d") were used to denote differences among green bean groups. However, we recognize that this distinction may not be immediately apparent to readers and could lead to misinterpretation. To address this concern, we have revised the figure legends to define the significance of uppercase and lowercase letters explicitly: p.5-6 Figure 1, p. 6-7 Table 1, p.7 Figure 2, p.8 Figure 3, and p.9-10 Figure 4.
These adjustments ensure that the rationale behind the notation is conveyed, minimizing the risk of confusion while maintaining the integrity of the data presentation. We sincerely appreciate your insightful suggestion.
[Comment] Line 306; "Costarica" is correct, not "Costa Rica"
[Responses] Thank you for carefully reviewing and identifying the typographical error. We sincerely apologize for this oversight and have corrected it to "Costarica," as you suggested. We greatly appreciate your attention to detail.
[Comment] Line 380, it is suggested to homogenize the text with the symbol "%", that is, the correct form is "25.55 and 40.88%"
[Responses] We sincerely appreciate your meticulous review and for pointing out the inconsistency in using the "%" symbol on line 380. In response, we have carefully reviewed the manuscript to ensure that it is consistently and correctly formatted throughout the text, including the example "25.55 and 40.88%" you highlighted: p.11 (lines 410-411, 413-414, 419, 431-433), p.12 (lines 446-447)
[Comment] Review, modify the scientific name in italics.
[Responses] Thank you for your thoughtful review and for bringing the formatting issue regarding the scientific names to our attention. In response, we have carefully reviewed the manuscript and ensured that all scientific names are correctly italicized throughout the text: p.2 (line 45) and references (12, 17, 29, 36, 39, 40, and 60).
We sincerely appreciate your attention to detail.
Round 2
Reviewer 1 Report
Comments and Suggestions for Authors
It is still not clear what is the point of calculating the significance of differences between the content of polyphenolic compounds in coffee leaves and coffee beans. What is this supposed to contribute to the work?
Author Response
It is still not clear what is the point of calculating the significance of differences between the content of polyphenolic compounds in coffee leaves and coffee beans. What is this supposed to contribute to the work?
We sincerely appreciate your thoughtful and constructive feedback.
Our study addresses a critical gap in the existing literature by adopting a novel and systematic approach. Previous studies have primarily focused on the effects of processing methods, such as roasting or drying, on the polyphenolic contents of a single part of the coffee plant [1] or have compared leaves and beans within a single cultivar [2] or across cultivars grown under differing environmental conditions [3], our research distinguishes itself by evaluating seven cultivars of Arabica coffee. All cultivars were grown in the same controlled environment and subjected to identical drying and extraction protocols, ensuring that any observed differences in polyphenolic composition are cultivar-specific and not influenced by external variables.
Our findings revealed significant variability among cultivars. While the majority exhibited higher total phenolic and flavonoid contents in leaves compared to beans, the Marsellesa cultivar displayed an opposite trend, with its beans containing higher levels of total flavonoids. This unexpected result challenges the commonly held assumption that coffee leaves universally contain higher levels of these compounds than beans. By demonstrating that these differences are cultivar-dependent, our study provides valuable insights into the variability of bioactive compound distribution within coffee plants.
This comparison is significant as it offers a foundation for targeted strategies to utilize coffee plant components in functional and sustainable applications. By confirming that polyphenolic content and distribution vary by cultivar, our results emphasize the importance of considering cultivar-specific characteristics in research and practical applications. These findings contribute to a deeper understanding of coffee plant bioactivity and its potential for value-added applications in food and nutraceutical industries.
We trust these revisions address your concerns and clearly articulate our work's contributions. Once again, we sincerely thank you for your insightful comments, which have helped us refine our discussion and enhance the impact of our study.
[1] S. Ngamsuk, T.-C. Huang, J.-L. Hsu, Determination of phenolic compounds, procyanidins, and antioxidant activity in processed Coffea arabica L. leaves, Foods 8(9) (2019) 389.
[2] E. Sadiyah, L. Purwanti, S. Hazar, S. Sasmita, A. Yuniarti, Total polyphenol and flavonoid content comparation of Kertasari Arabica coffee (coffea arabica L.) leaves, pulp, and beans, Medical Technology and Environmental Health, CRC Press2020, pp. 225-229.
[3] B. Mehari, B.S. Chandravanshi, M. Redi-Abshiro, S. Combrinck, R. McCrindle, M. Atlabachew, Polyphenol contents of green coffee beans from different regions of Ethiopia, International Journal of Food Properties 24(1) (2021) 17-27.